# Review of the Treatments for Central Neuropathic Pain

**DOI:** 10.3390/brainsci12121727

**Published:** 2022-12-16

**Authors:** Breanna L. Sheldon, Zachary T. Olmsted, Shelby Sabourin, Ehsaun Heydari, Tessa A. Harland, Julie G. Pilitsis

**Affiliations:** 1Department of Neuroscience and Experimental Therapeutics, Albany Medical College, Albany, NY 12208, USA; 2Department of Neurosurgery, Albany Medical College, Albany, NY 12208, USA; 3Department of Clinical Neuroscience, Charles E. Schmidt College of Medicine, Florida Atlantic University, Boca Raton, FL 33431, USA

**Keywords:** central neuropathic pain (CNP), pharmacological treatment, surgical treatment

## Abstract

Central neuropathic pain (CNP) affects millions worldwide, with an estimated prevalence of around 10% globally. Although there are a wide variety of treatment options available, due to the complex and multidimensional nature in which CNP arises and presents symptomatically, many patients still experience painful symptoms. Pharmaceutical, surgical, non-invasive, cognitive and combination treatment options offer a generalized starting point for alleviating symptoms; however, a more customized approach may provide greater benefit. Here, we comment on the current treatment options that exist for CNP and further suggest the need for additional research regarding the use of biomarkers to help individualize treatment options for patients.

## 1. Introduction

Central neuropathic pain (CNP) is defined as a neurological disorder resulting from damage to the central nervous system (CNS) sensory pathways including the brain, brainstem, and/or spinal cord [1]. Specifically, it is pain attributable to CNS lesions or diseases (not dysfunction) according to updated guidelines [2,3]. In CNP, the pain signals go awry via multiple mechanisms and persist to the point of maladaptation. For example, ectopic nociceptive signals may be erroneously generated, normal pain signals may be inappropriately modulated and/or sustained, or signals may be improperly propagated up to the brain [2,4]. The highly variable nature of the mechanisms causing symptoms contributes to difficulty treating CNP conditions. It is also important to distinguish CNP from central sensitization, or the hyperactivation of central nociceptive pathways leading to allodynia (where nonpainful stimuli such as touching a cheek are perceived as painful) and/or hyperalgesia (where painful stimuli produced perception of pain that is out of proportion) [3].

Diagnosing CNP can be difficult, as it is largely reliant on clinical symptoms that often present broadly. CNP is generally suspected in patients who complain of pain for more than 3 months following a brain injury or SCI, or who have concomitant disease (Multiple sclerosis (MS) or Parkinson’s Disease (PD), for example) with a negative laboratory workup. Symptoms may occur immediately following an injury or be delayed by days, months, or even years [1,4,5]. CNP may also be associated with cognitive disturbances, including but not limited to major depressive disorder, generalized anxiety disorder, fatigue, or other changes in mood or sleep [1]. It should also be noted that there is often considerable overlap between peripheral and central neuropathic pain, which can also co-occur in the same patient.

Comorbidities are common amongst patients suffering from CNP, and, as a result, the etiology of various complaints may be difficult or impossible to tease out. Other forms of pain, including musculoskeletal or peripheral neuropathic pain, should be ruled out or distinguished from CNP symptoms. Here, we comment on the current treatments for CNP and highlight areas of research interest moving forward.

## 2. Pathologies of CNP

CNP encompasses a broad array of ischemic, traumatic, metabolic, and neurodegenerative conditions affecting the CNS. Additionally, centralized pain can develop over time from peripheral pain secondary to chronic musculoskeletal and rheumatological syndromes.

Following cerebrovascular injury, central post-stroke pain (CPSP) can develop up to 12 months following the initial insult, with the incidence of pain onset varying depending on the location of the stroke. Recent studies suggest CPSP is most prevalent in patients with thalamic strokes, with up to 40% of patients experiencing CNP [6]. The mechanism of CPSP is not fully understood, but current theories suggest dysfunctional neuroplasticity within the spinothalamic pathway likely results in hyperexcitability and aberrant pain signaling.

Chronic pain following traumatic brain injury (TBI) or spinal cord injury (SCI) often has broader presentation, including headaches, peripheral pain, musculoskeletal pain, and central pain. The incidence of central pain following TBI is ambiguous due to the various mechanisms of trauma and interaction between molecular and psychosocial factors [7]. Notably, complaints of depression, PTSD, and sleep disturbances often co-occur with the development of neuropathic pain. Shearing forces during the initial insult are known to cause edema, diffuse axonal injury, ischemia, hypoxia, and excitotoxicity, the combination of which leads to varied gross and microscopic cerebral changes [7]. Specifically, changes in thalamic, pontine, cingulate, and prefrontal activity are thought to contribute to chronic pain due to altered supraspinal modulation of pain perception [8]. Similarly, in spinal cord injury (SCI), nociceptive transmission changes at the dorsal horn of the spinal cord are thought to contribute to hyperexcitability of cortical structures in the development of CNP [9]. Additionally, recent research suggests that reactive gliosis may contribute to the pathology of CNP in SCI [10]. However, these findings were not specific to SCI and have raised interest regarding the role of astrocytic activity in the pathology of CNP, regardless of traumatic, cerebrovascular, metabolic, or degenerative injury.

In neurological disease, such as MS and PD, the incidence of CNP is estimated to be up to 50% and 75%, respectively [11,12]. In the case of MS, the pathophysiology is likely unique compared to other causes of CNP due to the autoimmune nature of the disease. Similarly, dopamine imbalance in PD is thought to contribute to the development of central parkinsonian pain. Interestingly, there is growing speculation that centralized pain may be a unique subtype of PD that has been associated with compulsive behavior and younger age of disease onset [13]. Despite these differing causes, treatment of CNP is fairly consistent across etiologies, with pharmacologic management often being the first line of treatment.

## 3. Methodology of Review

Appropriate evidence for this narrative review was found utilizing electronic search databases, including PubMed, ScienceDirect, and ClinicalTrials. The literature review for this manuscript included articles from the past 32 years (1990–2022) that met the inclusion criteria. Such criteria included all types of articles published within the given timeframe, articles written in or translated into English, and research including participants with evidence of central pain. Articles were excluded if they met more than one of the following criteria: the article was withdrawn prior to this review, the research had previously been disproven within the scientific community, or if pharmacologic, surgical, or affective approaches to treating CNP were not mentioned. Keywords utilized to narrow the searching process included centralized pain, central neuropathic pain, chronic central pain, CNP, pharmacological treatment, surgical treatment. Selected articles were reviewed and integrated into this narrative review, as presented in the following sections.

## 4. Pharmacologic Treatments

Due to the nature of CNP, the resolution of symptoms, as opposed to the cause of the condition, is often the initial focus of treatment. As such, the first line of medical management is often conservative pharmacotherapy. Dissimilar to acute episodes of pain, CNP is often refractive to common analgesics, such as non-steroidal anti-inflammatory drugs [14]. Pharmacologic recommendations indicate antidepressants and anticonvulsants as first-line drugs, weak opioids and topical treatments as second-line drugs, stronger opioids and botulinum toxin A as third-line drugs, and nerve blocks when pain is refractory to standard medical management.

### 4.1. First-Line Pharmaceuticals

Antidepressants and anticonvulsants are consistently the most strongly recommended and well-studied pharmaceuticals in CNP (Table 1). The most recent systematic review across neuropathic pain diagnoses indicated similar analgesic success between serotonin–noradrenaline reuptake inhibitors (SNRIs), tricyclic antidepressants (TCAs), pregabalin, and gabapentin [15]. Interestingly, specific serotonin reuptake inhibitors (SSRIs) have received mixed recommendations regarding administration for CNP, with zimelidine and fluvoxamine having better results for central pathologies [16]. TCAs (most commonly amitriptyline) and SNRIs (most commonly duloxetine and venlafaxine) have shown notable efficacy amongst a range of neuropathic conditions [17]. Recent research suggests that TCAs and SNRIs have both central and peripheral mechanisms of action, though the precise mechanism is not fully clear. Centrally, these antidepressants act on α2 adrenergic receptors in the dorsal horn of the spinal cord, inhibiting signaling from primary sensory afferents and hyperpolarizing ascending neurons [18]. Peripherally, these agents are thought to act on β2 adrenergic receptors, resulting in delayed anti-inflammatory effects [19]. Conversely, anticonvulsants are thought to act on presynaptic α2δ subunits of voltage-gated calcium channels (VGCC) throughout the central and peripheral nervous systems, decreasing neurotransmitter release. Although the precise intracellular interaction between the anticonvulsants and the subunit are not well defined, gabapentin is believed to bind peripherally dominant α2δ1 subunits more efficiently than CNS-dominant α2δ2 and α2δ3 subunits [20]. This likely accounts for the more potent efficacy of anticonvulsants in peripheral neuropathies.

Given the analgesic success of these antidepressants and anticonvulsants separately, combination therapy was a promising concept. However, most studies indicate comparable relief with high-dosage monotherapy compared to lower dosage combination therapy [14]. Additional studies using weak opioids in combination therapy with antidepressants and anticonvulsants found similar efficacy to monotherapies [15]. Due to a lack of mechanistic studies regarding combination therapy, it is not well understood why the use of multiple agents does not elicit an additive analgesic effect. While these results are not indicative for combination therapy, they do implicate a treatment option for cases where high dosage pharmacotherapy is not ideal.

### 4.2. Second-Line Pharmaceuticals

Pharmacotherapies with mixed recommendations include weak opioids, topical capsaicin, lidocaine patches, and SSRIs. Tramadol is a non-traditional opioid that binds both mu opioid receptors, similarly to potent opioids, and prevents the reuptake of serotonin and norepinephrine within the CNS, similarly to SNRIs [15,21]. This characteristic contributes to both its beneficial and adverse effects. While the decreased affinity for opioid receptors lessens traditional side effects of opioids, such as tolerance, dependency, and respiratory changes, the effect on serotonin can incite complications with certain comorbidities [21]. With these caveats, tramadol has been recommended for a limited number of CNP pathologies [17]. Although originally receiving high recommendation for the medical management of peripheral neuropathies, recent indications suggest mild pain relief may be achieved with topical analgesics in cases of CNP associated with TBI [14,15,17]. Lidocaine patches are thought to act locally by blocking voltage-gated sodium channels and preventing aberrant neuronal firing in nociceptive transmission [14]. Capsaicin patch acts as a selective, local transient receptor potential cation channel subfamily V member 1 (TRPV1) agonist on nociceptive fibers, decreasing pain transmission rapidly [22]. An ongoing clinical trial is assessing the use of a combination of low-dose 0.025% and high-dose 8% patches in cases of SCIs refractory to anticonvulsant therapy (NCT02441660; Table 2). While SSRIs are less commonly prescribed for CNP as compared to other antidepressants, they are still occasionally considered, especially for the concomitant treatment of depression and pain. These agents inhibit serotonin reuptake from the synapse, prolonging synaptic transmission, but it is not well understood how this contributes to analgesic effects [16]. Given the potential varied side effect profile of these agents and the limiting efficacy of topical treatments in central pathologies, these pharmacotherapies receive moderate recommendation for treatment of CNP.

### 4.3. Additional Pharmaceutical Agents

Additional pharmacotherapies include traditional opioids and localized botulinum toxin A. Amongst full agonist opioids, morphine and oxycodone are most frequently utilized for CNP, though significant concerns including respiratory suppression, constipation, tolerance, dependence, and abuse have relegated them to last-line options [21]. Unlike their weaker counterpart, tramadol, these agonists have strong affinity for mu opioid receptors in addition to moderate interactions with delta (δ) and kappa (κ) opioid receptors [21]. While the precise mechanism of action of these agents in central pain analgesia is not fully understood, recent spinal and cortical EEG studies indicate that opiates likely act at both the spinal cord and cortical level to decrease nociceptive transmission [23]. The particularly high risk for hyperalgesia, and the extensive withdrawal period, warrants conservative prescription with close follow-up. While the diffuse effect of opioids renders them effective for most CNP conditions, localized botulinum toxin A injections are considered for cases of localized pain associated with SCI [15]. Classically, the neurotoxin inhibits acetylcholine release into neuromuscular junctions (NMJs) by preventing vesicular fusion with the presynaptic terminal and subsequent neurotransmitter release. However, in central pain, it is thought to decrease nociceptive transmission through axonal transport to supraspinal sensory nuclei, where the toxin mediates pain sensation [24]. Given that the treatment is restrictive to localized symptoms and requires specialist administration, it is not highly recommended in the treatment of most central pain pathologies.

### 4.4. Emerging Therapies and Ongoing Clinical Trials

The complex nature of CNP and variations in pharmacologic recommendations over the years encourage the continued pursuit of novel therapeutics. Recent studies have concentrated on antagonists of sodium channel subtype Na_v_1.7, which is found in both the CNS and peripherally and thought to be involved in the processing of neuropathic and inflammatory pain [25]. Given the association of aberrant glutamatergic NMDA activity with nociceptive processing, NMDA antagonists, most notably low dose ketamine, have also been evaluated with some success [26]. However, the cardiovascular and psychological risks associated with the drug class is an area of concern for its potential use. Recent clinical trials regarding the use of NMDA antagonists have been limited to cases of generalized neuropathic pain not restricted to central pathologies (NCT04459234). Similarly controversial is the use of cannabinoids, as the reports of moderate pain relief are often accompanied by safety concerns due to the drug’s influence on affect and sedation [15,17,26]. Synthetic cannabinoids, both delta 9-tetrahydrocannabinol (delta 9-THC) and cannabidiol, have received attention for varied reports of improvement in both pain and spasticity in patients with MS [27]. The current theory behind the analgesic effect of these agents centers around the targeting of affective components of pain in the frontal cortex and limbic system [28]. While current studies regarding the use of medicinal cannabinoids focus on the agent’s utility in improving peripheral pain syndromes, there is one ongoing clinical trial investigating the use of nabilone and dietary changes in the management of CNP following SCI (NCT04057456). Historically, the antiarrhythmic mexiletine received attention in managing thalamic pain following stroke [29]. The oral agent works similarly to lidocaine by inhibiting sodium channels in the dorsal horn and was found to have few and minor adverse effect. While few current studies include mexiletine, one recent clinical trial investigated the analgesic effect of quinidine, sodium channel blocker, and dextromethorphan, opioid agonist and NMDA receptor antagonist combination therapy in treating CNP in patients with MS (NCT01324232) [30].

## 5. Surgical Treatments for Central Pain

While pharmacologic regimens are generally first line in the treatment of CNP, each carry their own adverse effects and consequences that may limit their utility. Multidisciplinary care has been shown to reduce pain, improve mood, decrease pain catastrophizing, and increase pain acceptance by patients [31]. For these reasons, more invasive therapies are often employed in addition to medical therapies for treatment of CNP. Figure 1 depicts the approximate order in which treatments may be deployed, though exact regimens vary widely by specific indication.

### 5.1. Motor Cortex Stimulation

Motor cortex stimulation (MCS) involves electrodes placed in the epidural space over the primary motor cortex (precentral gyrus, M1) to provide chronic stimulation. The somewhat counterintuitive superiority of M1 as the stimulation target over the primary sensory cortex (postcentral gyrus) was discovered in initial studies by Tsubokawa et al. [32]. The mechanisms of MCS are incompletely understood, though animal models have implicated activation of interneurons in the periaqueductal gray leading to improved pain control [33,34]. Increased blood flow to the cingulate gyrus correlates with greater pain relief, therefore MCS may be modifying the affective component of pain [35]. Complications with MCS include those related to the surgery, including intracranial bleeding/hemorrhage, infections, and permanent neurologic disability [35,36]. Additionally, MCS can induce seizure activity, though this is almost always reversible.

### 5.2. Deep Brain Stimulation

Deep brain stimulation (DBS) is an additional intracranial stimulation option. Electrodes are implanted in the deep nuclei to modulate nociceptive signaling. The stimulation target varies based on indication, with the most common for pain being the periaqueductal gray, periventricular gray matter, and ventroposterolateral thalamus, though the nucleus accumbens has also shown potential [37,38,39,40]. Stimulation parameters often target multiple or all of these locations, with the analgesic effects thought to involve the release of endogenous opioids.

Initial forays into the use of DBS for chronic pain conditions were disappointing. Efficacy varies wildly depending on the indication, and initial prospective multicenter studies showed insufficient evidence to secure FDA approval for DBS as a treatment for chronic neuropathic pain [35,41,42,43]. However, subsequent studies suggest that DBS achieved significant pain reduction in 50–70% of central post-stroke pain (CPSP) patients, indicating the key to success may be deliberate and careful patient selection. DBS also poses risk of postoperative infections, bleeding, and permanent neurologic deficits [35,38]. Further, up to one-third of patients permanently implanted eventually have all their hardware removed due to various reasons [37].

### 5.3. Lesioning

Lesioning involves the irreversible destruction of nociceptive pain pathways, often called neuroablation. This technique is one of the oldest surgical options employed for pain and can be conducted via cingulotomy and thalamotomy. One can induce immediate relief from pain by permanently disrupting the nociceptive signaling pathways, though at least one study testing somatosensory sensations in patients who underwent lesioning for CNP found that the spinothalamic tract is not solely responsible for the pain sensations [44]. Central lateral thalamotomy via gamma knife surgery and focused ultrasound (FUS) has been shown to significantly improve outcomes.

While the instant pain relief promised by lesioning is tempting for many patients, there are several caveats to consider. In addition to normal surgical complications such as infection and hemorrhage, unpleasant adverse effects such as numbness, weakness, or even new-onset neuropathic pain may occur and the mechanism lacks the adjustability and reversibility of other treatment options [45]. Further, the relief obtained is often temporary, and lesioning’s eventual loss of efficacy and surgical risks limits its use [3,45,46].

### 5.4. Focused Ultrasound

FUS is a relatively new tool in the field of neuromodulation [47,48]. Using acoustic rather than electromagnetic energy, FUS can achieve stimulation of deep structures with high spatial resolution. A vast spectrum of frequencies and intensities enables a range of possible outcomes from reversible neural activity modulation (low-intensity, low-frequency) to irreversible ablation of neural tissue (high-intensity) such as for thalamotomy [49]. Jeanmonod and colleagues in Switzerland first achieved noninvasive ablation of the central lateral nucleus of the thalamus in chronic neuropathic pain patients to attenuate features of intractable pain [49]. This small study necessitated long-term clinical trials, with two currently ongoing (NCT03111277, NCT03309813). A global open registry of thalamotomy procedures has also been generated (NCT03100474). An interest exists in neuromodulatory, reversible low-intensity, low-frequency ultrasound applied to CNP treatments, but these efforts also remain in the early research stage [50,51,52].

### 5.5. Spinal Cord Stimulation

Spinal cord stimulation (SCS) is an invasive neuromodulatory technique that is used in patients with neuropathic pain. While SCS treats many peripheral pain conditions, its utility in CNP syndromes is unfortunately limited to last-line treatment due to findings that its efficacy inevitably wanes and pain almost always returns within a few years [3,53,54,55,56,57,58]. Pain caused by thoracic lesions tends to have better outcomes, though relief has also been achieved in patients with cervical lesions [53,54,57]. Analgesia is thought to be achieved by modulation of both ascending nociceptive signals and descending modulatory signals [59]. Further, SCIs caused by trauma often require emergent operations which may include hardware that preclude the use of SCS later down the road [3].

## 6. Alternative Therapies

While effective for many patients, invasive procedures introduce potential for complications as well as waning pain relief with time [60,61,62]. Many patients consider alternative interventional therapies either in lieu of or in combination with current standard options [63,64,65]. We therefore expand upon these options and their supporting evidence in the treatment of CNP below.

### 6.1. Noninvasive Central Stimulation

Neuronal properties can be modulated from outside of the intact cranium using electromagnetic transducers targeting specified brain regions. This phenomenon, known as transcranial cortical neurostimulation, is the basis for noninvasive stimulation of CNS compartments to modulate functional output and sensory perception [42]. Such modalities include repetitive transcranial magnetic stimulation (rTMS), transcranial direct current stimulation (tDCS), transcranial random noise stimulation (tRNS) and cranial electrotherapy stimulation (CES) [63,66]. These methods have been applied to a range of CNP syndromes including CPSP, post-SCI pain, and MS [65,67,68], although the efficacy of these therapies remains under investigation in the management of neuropathic pain.

### 6.2. Repetitive Transcranial Magnetic Stimulation

Noninvasive rTMS was shown to mitigate pain symptoms in patients with intractable CNP etiologies, producing similar analgesic effects to MCS with similar mechanisms [69,70]. Modulation of opioid, GABAergic, glutamatergic, and descending inhibitory pathways is thought to be involved [71,72]. rTMS is able to induce protracted periods of analgesia for days to weeks following a single stimulation session, an effect which is pronounced with repetitive sessions [73]. rTMS carries an associated risk of seizures that can be mitigated by tailoring the pulse train and intertrain durations of the stimulation program to within the safety limits [74].

## 7. Cognitive, Affective, and Emotional Approaches

The multimodal theory of pain includes critical cognitive and emotional features in addition to peripheral nociceptive physiology. In essence, pain is a subjective, conscious perception separable from pure nociception. Depression and even suicide is not uncommon in patients suffering from CNP, and factors of depression, loneliness, and anxiety have been associated with poorer outcomes following SCS [75,76,77,78]. However, these cognitive components also constitute unique avenues for pain management that can be applied alone or in addition to standard treatments to enhance recovery.

### 7.1. Mindfulness Meditation

Research expanding on the concepts of interoception, or perception of sensations from inside the body, has led to a more refined mechanistic understanding of conscious interoception and self-awareness [79]. With the goal of distinguishing one’s thoughts about pain from the physiologic experience of the pain itself, meditation-mediated stress reduction has been shown to alter brain activity on fMRI and reduce pain in multiple etiologies of chronic pain [80,81]. Further decoding the neural mechanisms between meditation and pain reduction is being actively pursued as an exciting, unfolding sector in pain management.

### 7.2. Cognitive Behavioral Therapy and Methods of Clinical Assessment

Similarly, cognitive behavioral therapy (CBT) techniques aim to reframe the patient’s own perception of pain. These techniques address mood, functioning, and social engagement. However, a number of systematic reviews of psychological intervention trials for chronic pain did not produce evidence for or against chronic pain management in terms of safety and efficacy [82,83,84]. A need therefore exists for psychological intervention trials specifically designed for CNP patients considering unique psychosocial disparities and disease-specific features including multiple sensory challenge, comorbidity, and polypharmacy.

## 8. Future Directions

Since chronic pain is multidimensional, involving sensory, cognitive, and affective components, each case of chronic pain is as individualized as the patient themselves. Personalized therapies require a systematic, customizable, stratified approach to ensure the most effective, tolerable, and safe treatment for their needs. Individualized plans will evolve closely with the identification of CNP biomarkers, which include omics, neuroimaging, electrophysiology, bioassay and behavioral markers [85]. Surgically, an advanced understanding of predictive factors of treatment responses directly related to the evolution of biomarkers will aid in determining where, when, and in whom invasive procedures would be appropriate for [86,87]. The integration of invasive platforms with closed-loop neural signature biomarker feedback when appropriate is expected to drive optimized, individualized pain management [88,89]. These frontiers that are already underway emphasize the expansive future of CNP management with infinite potential.

## 9. Conclusions

Though CNP remains debilitating to millions of people worldwide, many treatment options exist to help combat the symptoms and improve the lives of those suffering. While pharmaceuticals and surgical treatments classically represent the major weapons to fight these chronic pain conditions, some choose to pursue relief via alternative therapies, including noninvasive stimulation and behavioral interventions such as CBT and meditation. Further studies in genetics and biomarkers may both ease diagnosis of CNP as well as provide tailored treatment plans. With constant technological advances and flourishing research in the field, we inch towards treating pain, minimizing adverse effects, and bettering the lives of sufferers worldwide.

## Figures and Tables

**Figure 1 brainsci-12-01727-f001:**
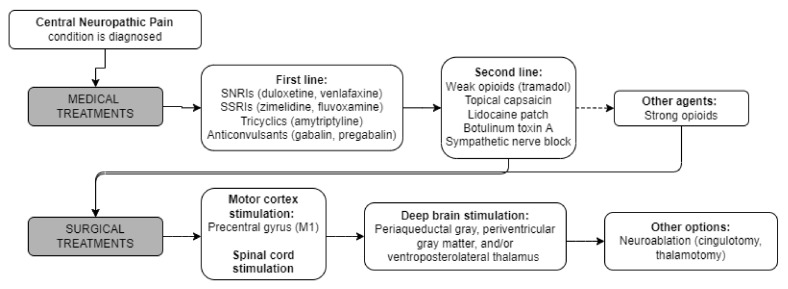
Flowchart of potential treatments for central neuropathic pain conditions.

**Table 1 brainsci-12-01727-t001:** Guideline of Pharmacotherapy Recommendations. Summary regarding pharmacologic recommendations for the treatment of centralized pain conditions.

Drug Class	Site of Action	Pharmacotherapy	Recommendation Strength	Dosage Ranges	Indications for Recommendation
**TCAs**	Increase α2 adrenergic transmission in the dorsal horn; increase activity at β2 receptors peripherally	Amitriptyline	Strong	25–150 mg/day	All conditions
**SNRIs**		Duloxetine	Strong	60–120 mg/day	All conditions
		Venlafaxine	Strong	150–225 mg/day	All conditions
**Anticonvulsants**	Decrease synaptic transmission by blocking presynaptic VGCC centrally and peripherally	Lamotrigine	Strong	200 mg/day	All conditions
		Pregabalin	Strong	300–600 mg/day *	Spinal cord injury (SCI), Traumatic brain injury (TBI), and Multiple sclerosis (MS)
		Gabapentin	Strong	1200–3600 mg/day *	SCI, TBI, and MS
**SSRIs**	Inhibit serotonin reuptake from the synapse	Zimelidine	Inconclusive	Inconclusive	All conditions
		Fluvoxamine	Inconclusive	Inconclusive	Central post-stroke pain (CPSP)
**Opioids**	Weakly agonize mu receptors; inhibit norepinephrine and serotonin reuptake	Tramadol	Moderate	200–400 mg/day *	TBI
	Potently agonize mu receptors, moderately agonize κ and δ receptors	Buprenorphine	Inconclusive	Inconclusive	Recommendation for those at risk of opioid use disorder
		Morphine	Weak	120–240 mg/day	All conditions
		Oxycodone	Weak	120–240 mg/day	All conditions
**Topical agents**	Selectively agonize TRPV1 channels on nociceptive fibers	Capsaicin 8% patch	Moderate	1–4 patches for 60 min every 3 months	TBI
	Inhibit voltage-gated sodium channels	Lidocaine patch	Moderate	1–3 patches for up to 12 h a day	TBI
**Neurotoxin**	Decrease neurotransmitter release at the NMJ and at supraspinal sensory nuclei	Botulinum toxin A	Moderate	50–200 units every 3 months	SCI or pathologies with localized symptoms

* indicates administration in 2–3 divided doses.

**Table 2 brainsci-12-01727-t002:** Ongoing Clinical Trials in CNP. Summary regarding ongoing variations in pharmacotherapy being assessed for the treatment of central pain pathologies.

Clinical Trial	Site of Action	Pharmacotherapy	Dosage	CNP Indications
**NCT02441660**	Selectively agonize TRPV1 channels on nociceptive fibers	8% capsaicin patches	Sequence of low-dose vs. high-dose topical patch administration	SCI
**NCT04057456**	Agonize endocannabinoid receptors	Nabilone	0.5 mg nabilone +/− anti-inflammatory diet	SCI
**NCT04459234**	Antagonize NMDA receptors	Ketamine	Not disclosed	General neuropathic pain
**NCT01324232**	Weakly agonize opioid receptors and antagonize NMDA receptors/Inhibit sodium channels	Dextromethorphan/quinidine	20–45 mg dextromethorphan/10 mg quinidine	MS

## Data Availability

Not applicable.

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
