# Peer review of "Review of the Treatments for Central Neuropathic Pain"

_brainsci, 2022, doi:10.3390/brainsci12121727_

Round 1

Reviewer 1 Report

Dear Authors,

I very much like the topic of your review paper and see benefit to a review explaining treatments for CNP. The writing is good and you are off to a great start with your references.

However, this paper is not identified as a particular type of review. In my opinion this paper would be better identified as a Scoping Review, yet this paper is missing essential features such as;

1) Defining what type of review it is

2) Detailed methods section describing search strategy from at least 1-2 search databases, period of literature search, inclusion and exclusion criteria, article screen process etc....

3) More recently reviews (even non systematic reviews) are registered prior to completion of the study, usually open science framework or other.

4) There are no guidelines followed such as PRISMA and the extension for scoping reviews. Some papers even follow PRISM-ScR framework as an outline for completing a narrative review. This is essential to be considered.

Without a detailed and clearly laid out search strategy a review like this one does not truly offer a novel contribution to the literature. Having said that, you have written a nice review, that if supported by detailed and guidelines based search strategy would likely be worthy of publication.

Either this work could be considered an opinion piece and restructured and condensed, or it needs to be redone. 

Reviewer 2 Report

Sheldon et al attempt to provide a comprehensive review of the therapeutics for treating central neuropathic pain. The review topic is interesting and would be helpful to scientists in the field, and medical practitioners. However, a few things need to be improvised before acceptance.

I have three major points:

1.       Types of Central neuropathic pain should be discussed as an independent section.

2.       There should be a more elaborate discussion of the results of clinical studies for the mentioned drug therapies.

3.       A table for clinical trials that are completed or ongoing.

Other comments

1.       “Central neuropathic pain (CNP) affects millions worldwide” Please provide some relevant statistics at least for the US

2.       “use of biomarkers to help individualize treatment options for patients.” Part of the review that discusses the biomarkers does not provide any strong evidence of biomarkers that can be used. Davis et al have talked about the use of biomarkers in terms of their required characteristics and potential use. So I would recommend removing that sentence from the abstract because it can be misleading to the reader’s expectations

3.       Please delete this. The information seems to be unhelpful in the introduction to central neuropathic pain. “Pain in response to injury can be 24 adaptive. For example, if one sprains one’s ankle, painful twinges felt while walking re- 25 mind one to minimize further damage and permit the appendage time to heal and re- 26 cover”

4.       Please change the word upregulation to hyperactivation “It is also important to distinguish CNP from central sensitization, or the upregulation of”

5.       Please rephrase this sentence “pathic pain and CNP in the same patient and even within the same condition” Not cleasr what author wants to communicate by writing even within the same condition.

6.       Topical Capsaisin patch therapy should be discussed a little more as it is under clinical trial with proper citation.

7.       Please discuss Mexiletine for thalamic pains syndromes (Awerbuch et al 1990)

Reviewer 3 Report

Here, Sheldon et al. reviewed and evaluated current therapeutic strategies for curing central neuropathic pain. The pain therapies they summarized in the manuscript included the major existent treatments for neuropathic pain, including pharmacological, surgical as well as noninvasive methods. Within the manuscript, the authors clearly stated the advantages, limitations as well as side effects of each therapy, and also introduced the sequence and priority for patients with distinct pain level to choose.

However, one major issue of this review is its lack of mechanistic analysis and comments of each therapeutic strategy of pain, which address the molecular, cellular or circuitry mechanisms of pain suppression and clarify the reason of occurrence of different side effects. For instance, in the section of 2.1, the authors commented the application of antidepressants could enhance norepinepherine activity through spinal adrenergic receptors, trying to explain they inhibited nociceptive transmission at peripheral sensory neurons and ascending projection neurons. However, it is still unclear what cell types in the periphery and spinal cord were affected by application of antidepressants. Also, why did combination therapy not work well? What's the potential molecular mechanisms? These questions might not be completely understood at the moment, but it would still be very important for the authors to give any comments or even just hypotheses. Similar case also apply to other pain therapies as well. 

In the Table 1, it will be necessary to list all the molecular and cellular targets of each drug class. For example, Tricyclic antidepressants, Targets-a2 adrenergic receptor-expressing spinal neurons. 

Round 2

Reviewer 1 Report

I think the additions you made to the paper are good. However, I do not think this is ready yet. Please list how many papers you identified in your search and excluded. Since you used PRISMA guidelines you may want to do a flow diagram. I would further recommend making another table to outline the studies you found. For example, list the studies you used to comment on "Pharmaceutical agents", list the author and year and what type of paper it was, and cite each study in the table.

Even Narrative reviews in recent years will list the search hits that were collected from each topic or each database.

For example take a look at this review that may serve as a model for the type of detail you should include even in a narrative review.( it is open access).

Each section of this study lists the number of search hits and relevant papers.

Fernández-de-Las-Peñas C, Florencio LL, Plaza-Manzano G, Arias-Buría JL. Clinical Reasoning Behind Non-Pharmacological Interventions for the Management of Headaches: A Narrative Literature Review. Int J Environ Res Public Health. 2020 Jun 9;17(11):4126. doi: 10.3390/ijerph17114126. PMID: 32527071; PMCID: PMC7312657.

Author Response

Thank you for your recommendations. We agree that PRISMA guidelines are important for systemic reviews and meta-analyses to summarize the filtering and selection process of articles. While PRISMA guidelines can be relevant for narrative reviews, we do not believe we can effectively apply it here as a large portion of the included articles were reviews. We have excluded all mention of PRISMA guidelines. If the reviewer feels strongly, we can go back and rerun the search by strict guidelines to ensure that we are more transparent on the process.

Reviewer 2 Report

I have no suggestions to make at this point. 

Author Response

Thank you.